# Multi-Temporal Remote Sensing Image Matching Based on Multi-Perception and Enhanced Feature Descriptors

**DOI:** 10.3390/s25175581

**Published:** 2025-09-07

**Authors:** Jinming Zhang, Wenqian Zang, Xiaomin Tian

**Affiliations:** 1School of Remote Sensing and Information Engineering, North China Institute of Aerospace Engineering, Langfang 065000, China; zhangjm1129@163.com; 2Aerospace Information Research Institute, Chinese Academy of Sciences, Beijing 100094, China; zangwq@aircas.ac.cn

**Keywords:** image matching, multi-temporal remote sensing images, deep learning, descriptor enhancement

## Abstract

Multi-temporal remote sensing image matching plays a crucial role in tasks such as detecting changes in urban buildings, monitoring agriculture, and assessing ecological dynamics. Due to temporal variations in images, significant changes in land features can lead to low accuracy or even failure when matching results. To address these challenges, in this study, a remote sensing image matching framework is proposed based on multi-perception and enhanced feature description. Specifically, the framework consists of two core components: a feature extraction network that integrates multiple perceptions and a feature descriptor enhancement module. The designed feature extraction network effectively focuses on key regions while leveraging depthwise separable convolutions to capture local features at different scales, thereby improving the detection capabilities of feature points. Furthermore, the feature descriptor enhancement module optimizes feature point descriptors through self-enhancement and cross-enhancement phases. The enhanced descriptors not only extract the geometric information of the feature points but also integrate global contextual information. Experimental results demonstrate that, compared to existing remote sensing image matching methods, our approach maintains a strong matching performance under conditions of angular and scale variation.

## 1. Introduction

With the rapid advancement of remote sensing technology, multi-temporal remote sensing images are playing an increasingly important role in areas such as ecological environment monitoring, land cover change detection, and disaster assessment [1,2]. However, significant geometric and radiometric inconsistencies exist between remote sensing images captured at various imaging times, sensor parameters, atmospheric conditions, and changes in the features themselves [3,4]. These inconsistencies pose significant challenges to achieving high-precision image matching. Reliable image matching is fundamental to multi-temporal analysis, as its accuracy directly impacts the precision of subsequent applications such as change detection and temporal modeling [5,6].

Traditional image matching methods face numerous challenges during multi-temporal image matching, including insufficient feature point counts, high rates of mismatch, and inadequate adaptability to nonlinear deformations [7,8]. In recent years, deep learning approaches, such as convolutional neural networks and attention mechanisms, have demonstrated significant potential in the field of image matching, enhancing matching robustness by learning high-level semantic features [9,10]. However, existing methods still require optimization regarding feature representation and matching stability, particularly the complex changing characteristics of multi-temporal remote sensing images. On the one hand, the two remote sensing images to be matched are acquired at different times, resulting in potential changes in the features they depict. This results in a reduction in the areas that can be effectively matched, making it challenging to achieve the desired quantity and accuracy of extracted feature points. On the other hand, multi-temporal remote sensing images are influenced by factors such as lighting conditions and imaging angles, often exhibiting significant radiometric differences and geometric distortions. These variations complicate the process of achieving stable and accurate matches when image feature points undergo descriptor matching.

To address these limitations, inspired by the efficient feature point detection paradigm of SuperPoint [11] and the core idea of feature enhancement in FeatureBooster [12], we propose a multi-temporal remote sensing image matching framework based on multi-perception and feature description enhancement. Unlike SuperPoint, which relies on a single visual texture to generate feature points, our framework’s feature extraction network is designed with multiple perception layers that focus on multidimensional remote sensing features. Furthermore, although numerous descriptors have been proposed in the field of image matching to meet the demands of multi-temporal scenarios, little attention has been given to enhancing existing descriptors, particularly through learning-based methods. Therefore, our approach innovatively optimizes the remote sensing image matching process from two aspects: Feature point extraction and feature descriptor enhancement. This method includes a feature extraction network that integrates multiple perception layers and a descriptor enhancement module designed to improve the performance of feature descriptors. The feature extraction network incorporates multiple perception layers, allowing the model to increase its focus on critical feature regions and enhance the distinctiveness of features. Additionally, multi-perception layers combine attention mechanisms with depthwise separable convolutions, allowing for independent feature extraction across channels and inter-channel feature fusion. This approach preserves local feature information, generates richer feature representations, and can reduce computational load while improving overall model performance to some extent. Furthermore, the descriptor enhancement module optimizes the feature descriptors through self-enhancement and cross-enhancement stages to improve their matching performance. The self-enhancement stage involves a multilayer perceptron (*MLP*) that encodes the geometric attributes of key points and combines this information with new descriptors. The cross-enhancement stage leverages a lightweight Transformer to capture spatial contextual cues, further enhancing the discriminative power and robustness of the descriptors.

The main contributions of this paper are as follows:We designed a feature extraction network based on multi-perception, which increases the number of effective feature points extracted in multi-temporal remote sensing image matching tasks and improves the positional accuracy of feature points to some extent.We introduced a feature descriptor enhancement module for multi-temporal remote sensing image matching tasks. This module optimizes feature descriptors through self-enhancement and cross-enhancement phases to improve their matching performance.To demonstrate the superiority of our method, we conducted comparative experiments against existing remote sensing image matching methods on multi-temporal remote sensing datasets, validating the accuracy and robustness of our approach.

## 2. Related Works

Existing remote sensing image matching methods can be categorized into two main types: region-based and feature-based approaches [13,14,15]. Region-based methods assess similarity using shallow information such as image intensity or phase [16]. However, the application of these methods is limited due to the susceptibility of pixel values to noise interference in remote sensing images. By contrast, feature-based matching methods focus on prominent features within the images, such as points, lines, or surfaces [17]. Matching based on image features is more effective at resisting the effects of noise and intensity variations, as well as adapting to deformations and changes in lighting, making it more suitable for remote sensing image matching in multi-temporal scenarios.

Traditional feature matching algorithms, such as SIFT [18] and ORB [19], are widely utilized to match various types of remote sensing images. However, these conventional methods face several limitations in multi-temporal remote sensing image matching. These include deficiencies in feature stability under dynamic object interference, as well as insufficient sensitivity to radiometric changes and geometric deformations [20,21,22]. Consequently, the accuracy and robustness of matching are significantly compromised.

With the rapid advancement of deep learning technologies, numerous deep learning-based feature matching methods have been applied to multi-temporal remote sensing image matching. Among these, SuperPoint [11] is a self-supervised dual-branch network model that simultaneously extracts the locations of feature points and generates descriptors. LoFTR [23] is a Transformer-based model for local feature matching that enhances matching accuracy in weak texture scenarios through a detector-free design. However, in complex multi-temporal environments, remote sensing image matching may encounter issues such as low matching accuracy or failure. To address these challenges, we propose a novel framework for multi-temporal remote sensing image matching that optimizes the matching process through feature point extraction and descriptor enhancement.

## 3. Method

### 3.1. Overview of the Proposed Method

Our framework incorporates a feature extraction network that integrates multiple modalities of perception and a descriptor enhancement module designed to improve the performance of feature point descriptors. The main workflow and details are illustrated in Figure 1.

Initially, the encoder of the feature extraction network extracts high-level feature representations from the remote sensing imagery input, producing a feature tensor. Subsequently, the multi-modal perception layer further processes the output feature tensor from the encoder to enhance the focus on significant features (Figure 1a). Next, the feature decoder, comprising both the feature point detection decoder and the descriptor decoder (Figure 1b), identifies the locations of image feature points based on the input feature tensor while generating descriptors for each detected feature point. Finally, the extracted feature point descriptors are fed into the descriptor enhancement module (Figure 1c), which enhances the performance of these descriptors through self-enhancement and cross-enhancement phases. This leads to the matching of enhanced feature point descriptors, resulting in the final image matching outcomes.

### 3.2. Feature Extraction Network

(1) Shared Encoder: The purpose of the shared encoder is to reduce the dimensionality of the input images and extract low-dimensional yet information-rich feature representations. This encoder takes the form of a VGG-style architecture, consisting of convolutional layers, max pooling layers, and nonlinear activation functions. The input image first passes through a 3 × 3 convolutional layer, mapping it into a 64-dimensional feature space, followed by the application of the ReLU activation function to introduce nonlinearity. Subsequently, the image undergoes three convolutional stages, each comprising two 3 × 3 convolutional layers, with a ReLU activation function applied after each convolutional layer. After the convolutional layers are passed through in each stage, a 2 × 2 max pooling layer is employed for downsampling, effectively halving the size of the feature maps. Specifically, the first stage maintains 64 channels, extracting basic features before pooling to reduce the dimensions. The second stage increases the number of channels to 128, capturing mid-level features before further pooling. Finally, the third stage increases the channels to 256, focusing on high-level features before completing the third pooling operation. Following these three downsampling operations, an additional 3 × 3 convolutional layer reduces the number of channels from 256 to 128, serving as a shared feature input for subsequent key point detection and descriptor generation. Ultimately, the spatial dimensions of the images are reduced to one-eighth of their original size. If the input image has dimensions H × W, the output feature map will measure H/8 × W/8.

(2) Multi-Perspective Layer: To enhance the number of feature points in multi-temporal images and achieve higher positional accuracy, a multi-perspective layer is introduced into the feature extraction network, which combines attention mechanisms [24] with depthwise separable convolutions [25]. The main structure is illustrated in Figure 2.

Initially, the input feature map F passes through a channel attention module, where global average pooling and global max pooling compress the spatial information of each channel into two vectors. These vectors are then fed into a shared multi-layer perceptron (*MLP*). The *MLP* employs a two-layer fully connected network structure, using the ReLU activation function after the first fully connected layer, with the hidden layer dimension set to 1/16 of the feature map’s channel count. After processing using the *MLP*, the two vectors are summed and passed through a sigmoid activation function to generate channel attention weights $M_{c}$ (Equation (1)). Subsequently, this weight is multiplied with the original feature map F channel-wise to obtain the channel-weighted feature map $F^{\prime }$ (Equation (2)).(1)$$M_{c}(F)=\sigma (MLP(AvgPool(F))+MLP(MaxPool(F)))$$(2)$$F^{\prime }=M_{c}\left(F\right)⊗F$$

Here, σ denotes the sigmoid function; ⊗ denotes element-wise multiplication. During multiplication, the attention values are broadcast accordingly: channel attention values are broadcast along the spatial dimension and vice versa.

Subsequently, the feature map $F^{\prime }$ is processed by a spatial attention module. First, maximum pooling and average pooling are performed along the channel dimension to obtain two single-channel feature maps. These maps are concatenated and passed through a convolutional layer to reduce them to a single-channel feature map. After this, the sigmoid activation function generates the spatial attention weights $M_{s}$ (Equation (3)). Finally, this weight is multiplied pixel-wise with the channel-weighted feature map $F^{\prime }$ to produce the final enhanced feature map F″ (Equation (4)).(3)$$M_{s}\left(F^{\prime }\right)=\sigma \left(f\left(\left[AvgPool\left(F^{\prime }\right);\;MaxPool\left(F^{\prime }\right)\right]\right)\right)$$(4)$$F^{\prime \prime }=M_{s}\left(F^{\prime }\right)⊗F^{\prime }$$

Here, σ denotes the sigmoid function; f represents the convolution operation; and ⊗ denotes element-wise multiplication. During multiplication, the attention values are broadcast accordingly: channel attention values are broadcast along the spatial dimension and vice versa.

Moreover, the spatial attention module employs depthwise separable convolutions to aggregate the resulting feature representations. Specifically, a depthwise convolution is first applied to each channel along the spatial dimension, allowing for the retention of more local features within the channels. This approach is particularly effective for processing feature maps with complex scenes, as depthwise convolution can better capture characteristics such as the edges and textures of objects [26]. Subsequently, a 1 × 1 pointwise convolution is used to fuse the results of the depthwise convolutions. Through pointwise convolution, features from different channels can be effectively combined, enhancing the expressiveness of the features while reducing both the number of parameters and computational complexity, thereby improving the overall performance of the model [27,28].

(3) Key Point Detection Decoder: The feature point detection decoder first generates a feature tensor, where 64 channels represent the probabilities of feature points, and the 65th channel serves as an invalid channel (dustbin). Specifically, the first 64 channels correspond to the 64 possible sub-pixel locations within each 8 × 8-pixel block of the input image, indicating the candidate probabilities for each position to become a feature point. The 65th channel serves as an invalid channel, denoting that there are no valid feature points within the 8 × 8-pixel block. Subsequently, the Softmax function is applied to compute the probability distribution over the 65 channel values for each 8 × 8-pixel block, resulting in the probabilities for each sub-pixel location (first 64 channels) and the dustbin (65th channel). The probability distribution for each position can be formulated as follows:(5)$$P_{i,j}\left(k\right)=\frac{exp(F_{i,j}(k))}{\sum_{c=1}^{65}{exp(F}_{i,j}(c))}$$
where $P_{i,j}\left(k\right)$ represents the probability value of the k channel at the feature map position (i,j); $F_{i,j}\left(k\right)$ denotes the value of the input tensor at position (i,j) for channel k; $\sum_{c=1}^{65}{exp(F}_{i,j}(c))$ represents the summation over all channels c from 1 to 65; and exp() denotes the exponential function, which is used to convert scores into positive values in preparation for normalization.

For each 8 × 8-pixel block, the sub-pixel location with the highest probability among the first 64 channels is selected as a candidate point. If the probability of this location exceeds 0.01 and is greater than the probability of the dustbin channel, the candidate point is retained; otherwise, it is considered that the block contains no feature points. For the filtered candidate points, a non-maximum suppression (NMS) algorithm is performed with a radius of 3 pixels at the original image scale, examining all candidate points. For each candidate point, if its probability is the maximum within a 3 × 3 neighborhood, it is retained; otherwise, it is discarded to eliminate redundant points that are too spatially close.

(4) Feature Descriptor Decoder: The feature descriptor decoder generates a descriptor tensor, which is subject to L2 normalization for enhanced stability. Subsequently, bilinear interpolation is employed to sample the descriptor tensor at the locations of the key points, yielding a descriptor for each identified key point.

### 3.3. Descriptor Enhancement Module

To address the issue of poor stability for the feature point descriptors in multi-temporal remote sensing image matching, we introduced a descriptor enhancer. The model structure is illustrated in Figure 3 and primarily consists of two components.

(1)Self-Enhancement Phase: For each feature point detected in the image, a multilayer perceptron (*MLP*) network is applied to project its original descriptor; it is then mapped into a new space to obtain a preliminary enhanced descriptor. The *MLP* serves as a universal function approximator, as demonstrated by the Cybenko theorem [29]. We utilized *MLP* to approximate the projection function, denoting it as ${MLP}_{desc}$. The architecture consists of two fully connected layers, with an ReLU activation function applied between them. Additionally, layer normalization and dropout were utilized as regularization techniques to prevent overfitting. The transformed descriptor $d_{i}^{tr}$ for key point i was defined as the nonlinear projection of the extracted descriptor $d_{i}$:



(6)
$$d_{i}^{tr}\leftarrow {MLP}_{desc}\left(d_{i}\right)$$



To leverage the geometric information of feature points valuable for image matching, we employed another *MLP* $\left({MLP}_{geo}\right)$ to embed the geometric information into a high-dimensional vector, further enhancing the descriptor. We not only encoded the 2D position of the key point $\left(x_{i},y_{i}\right)$, but also included other relevant information such as scale $s_{i}$, orientation $\theta_{i}$, and detection score $c_{i}$. The high-dimensional embedding of the geometric information was added to the transformed descriptor as follows:(7)$$d_{i}^{tr}\leftarrow d_{i}^{tr}+{MLP}_{geo}\left(p_{i}\right)$$
where $p_{i}=(x_{i},y_{i},s_{i},\theta_{i},{\;c}_{i})$ represents all available geometric information as aforementioned.

From this, the descriptors processed through the *MLP* incorporate geometric information of the key points, resulting in enhanced discriminative power and robustness.

(2)Cross-Enhancement Phase: The self-enhancement phase does not consider the potential correlations among different feature points, enhancing each descriptor independently. For instance, it fails to leverage the spatial relationships between these feature points; however, spatial contextual cues can significantly improve feature matching capabilities. Consequently, the descriptors obtained after the self-enhancement phase perform poorly in scenarios involving variations in perspective and complex terrain. To address this issue, we further processed these descriptors through the cross-enhancement phase. We took the descriptors generated in the self-enhancement phase as input and utilized the attention mechanism of a Transformer to capture the spatial context relationships among the key points. We denote the Transformer as “Trans,” and the projection is described as follows:

(8)$$\left(d_{1}^{tr},d_{2}^{tr},\ldots d_{N}^{tr}\right)\leftarrow Trans\left(d_{1}^{tr},d_{2}^{tr},\ldots d_{N}^{tr}\right)$$
where the input of the Transformer is N local features within the same image, and the output comprises enhanced feature descriptors.

Furthermore, we employed an efficient Attention-Free Transformer (AFT) [30] to replace the multi-head self-attention (MHA) operation with the Vanilla Transformer. Unlike the traditional MHA architecture, AFT does not utilize or approximate dot-product attention. Specifically, AFT rearranges the computation order of Q (Query), K (Key), and V (Value), similar to linear attention, but multiplies the elements of K and V instead of using matrix multiplication. The no-attention Transformer for key point i can be formulated as follows:(9)$$f_{i}(X)=\sigma \left(Q_{i}\right)⊗\frac{\sum_{j=1}^{N}\exp\left(K_{j}\right){⊗V}_{j}}{\exp\left(K_{j}\right)}$$
where σ is a sigmoid function; $Q_{i}$ represents i-th row of Q; $K_{j}$, $V_{j}$ represent the j-th rows of K, V; ⊗ denotes element-wise multiplication; and exp() denotes the exponential function.

The Transformer can integrate global contextual information [31,32,33], allowing each descriptor to be adjusted based on its neighboring information, thereby further enhancing its discriminative power and robustness. The ultimately enhanced descriptors not only incorporate geometric information of the key points themselves but also fuse global contextual information, enabling them to handle challenging situations better, such as variations in perspective and local deformations.

(3)Loss Function: We used the descriptor matching problem as a nearest neighbor retrieval task and Average Precision (AP) to train the descriptors. Given the transformed local feature descriptor $d^{tr}=\left(d_{1}^{tr},d_{2}^{tr},\ldots d_{N}^{tr}\right)$, our objective was to maximize the AP [34] of all descriptors. Therefore, the training goal was to minimize the following cost function:



(10)
$$L_{AP}=1-\frac{1}{N}(\sum_{I}^{N}AP{(d}_{i}^{tr}))$$



To ensure that the original descriptors were enhanced, we proposed using an alternative loss function to compel the performance of the transformed descriptors to surpass that of the original descriptors:(11)$$L_{Enhance}=\frac{1}{N}\sum_{I}^{N}max(0,\frac{AP(d_{i})}{AP{(d}_{i}^{tr})}-1)$$

The final loss was the sum of the two aforementioned losses:(12)$$L=L_{AP}+{\lambda L}_{Enhance}$$
where λ is a weight to regulate the second term. We used a differentiable approach [35] to compute the Average Precision (AP) for each descriptor.

## 4. Experimental Results and Discussion

### 4.1. Experimental Data and Parameter Settings

In this study, we utilized two datasets: the LEVIR-CD dataset [36] and the CLCD dataset [37]. The LEVIR-CD dataset comprises 637 pairs of dual-temporal images spanning 5 to 14 years, with a spatial resolution of 0.5 m. It draws on information from various types of buildings, such as villas, high-rise apartments, small garages, and large warehouses. From the image pairs in the LEVIR-CD dataset, we selected 445 pairs for training, 128 pairs for validation, and 64 pairs for testing. The CLCD dataset comprises 600 pairs of cultivated land change sample images, captured by the GaoFen-2 satellite in Guangdong Province, China, during 2017 and 2019. The spatial resolution of these images ranges from 0.5 to 2.0 m and incorporates cultivated land, forest, shrubland, and grassland. For the CLCD dataset, we selected 360 pairs for training, 120 pairs for validation, and 120 pairs for testing.

All algorithms in this study were implemented using the PyTorch 1.8.0 framework on an Nvidia GeForce RTX 3090 GPU (Nvidia, Santa Clara, CA, USA). All experiments were conducted with a random seed set to 42. The optimizer used was Adam, with an initial learning rate of 10^−3^ and a minimum learning rate set to 10^−6^. The ε-neighborhood was set to three pixels. During training, the batch size was set to two, and the maximum number of training epochs was set to ninety. The learning rate was decayed every 10 epochs by multiplying it by a decay factor of gamma = 0.5, and the decay stopped when the learning rate reached the minimum threshold.

### 4.2. Performance Metrics

In the matching experiments, we used the following evaluation metrics for the algorithms: the number of correctly matched points (NCM), matching success rate (SR), root mean square error of matched points (RMSE), and runtime (RT).
(1)NCM: The NCM represents the total number of correctly matched points. One image in the pair is designated as the reference image, while the other serves as the target image. The feature point positions obtained from the matching algorithm on the target image are denoted as $x_{i}$, and the corresponding ground truth positions on the reference image are denoted as ${\bar{x}}_{i}$. Using a matching accuracy threshold ε (in this experiment, points within a 3-pixel error margin were considered correctly matched), the determination of correctly matched points is shown in the following equation:(13)$$∥H\left(x_{i}\right)-{\bar{x}}_{i}∥\;\leq \varepsilon$$
where H() is the mapping matrix between the reference image and the matched image.(2)SR: The SR is defined as the ratio of the number of correctly matched points (NCMs) to the total number of matched points provided by the algorithm (N). The calculation formula is as follows: (14)$$SR=\frac{n}{N}$$
where n represents the NCM.(3)RMSE: The RMSE is a commonly used metric for evaluating the performance of matching algorithms. It measures the degree of deviation between the predicted positions of matched points and their corresponding ground truth values. Specifically, the RMSE represents the average deviation of the matching points, reflecting the magnitude of the error at each matched point coordinate. The calculation formula is as follows:(15)$$RMSE=\frac{1}{n}\sum_{i=1}^{n}∥H\left(x_{i}\right)-{\bar{x}}_{i}∥$$
where n represents the NCM and H() represents the mapping matrix between the reference image and the matched image.

### 4.3. Matching Performance Evaluation

To comprehensively evaluate the image matching performance of our proposed method, we conducted a comparative analysis with several state-of-the-art image matching algorithms, including SIFT [18], SOSNet [38], D2-Net [39], SuperPoint [11], and LoFTR [23]. All methods were assessed using a publicly available code and the optimal parameter settings provided by the respective authors. The datasets and training configurations were consistent with those utilized in this research. We evaluated our model using the test sets selected from the LEVIR-CD and CLCD datasets, which comprise three groups of image pairs:(1)The original dual-temporal remote sensing image pairs;(2)Dual-temporal remote sensing image pairs subjected to angular transformations;(3)Dual-temporal remote sensing image pairs subjected to scale transformations.

Specifically, we conducted tests under multiple parameters, including rotation angles ranging from 10° to 40° and scaling factors from 0.5 to 0.9. For demonstration and analysis, we selected image pairs corresponding to a rotation angle of 30° and a scaling factor of 0.7 as representative examples. For each test, image matching was performed using a matching error threshold of 3 pixels. The results obtained from experiments conducted on the test set were averaged. Figure 4 presents representative image pairs demonstrating the matching results of different algorithms, while Table 1 summarizes the average quantitative evaluation results of six methods across three groups of test datasets.

In the first group, the LEVIR-CD dataset serves as a high-resolution building change detection dataset, with challenges primarily arising from occlusion and repetitive textures. As a representative of traditional algorithms, SIFT performs poorly compared to other learning-based algorithms across various metrics, mainly due to its reliance on handcrafted feature extraction processes, which often fail to match or result in inaccuracies when changes in land cover exceed expected ranges. SOSNet optimizes local descriptor learning through second-order similarity regularization, demonstrating good performance in simple building regions; however, it suffers from mis-matching due to similar textures and weak feature discrimination. Both D2-Net and SuperPoint algorithms exhibit overall strong performance, with success rates (SRs) exceeding 90%. The matching time of the SuperPoint algorithm is notably the shortest at 0.76 s, which is attributable to its end-to-end output descriptors generated by a fully convolutional network, leveraging high GPU parallel computing efficiency. LoFTR effectively addresses challenges related to occlusion and fragmented boundary matching via global attention and Transformers, and achieves excellent performance across all metrics, with a root mean square error (RMSE) of just 1.58. Our method achieved the highest number of correctly matched points (NCM) and SR, reaching 263 and 99.25%, respectively. This indicates that our method provides a rich quantity of feature points while maintaining high accuracy among the successfully matched feature points. Although our RMSE is higher than that of LoFTR, our method outperforms it in inference speed. For the CLCD land cover dataset, the primary challenges include mixed pixels and seasonal spectral differences. All algorithms performed worse on the CLCD dataset compared to the LEVIR-CD dataset, primarily because the LEVIR-CD dataset contains predominantly well-formed building areas and has a higher resolution than the CLCD dataset. Our method outperformed other algorithms in terms of NCM, SR, and RMSE on the CLCD dataset, although the matching time was longer than that of the SuperPoint algorithm. This can be attributed to our method’s design, which is suitable for images that are multi-temporal or that have temporal differences by utilizing a multi-sensing fusion feature extraction network to enhance feature discrimination capability, while also optimizing the performance of feature point descriptors through a descriptor enhancement module, albeit at the cost of increased network complexity. Comparisons on the LEVIR-CD and CLCD datasets demonstrate that our method provides accurate matching in scenarios involving land cover changes and seasonal lighting differences, while maintaining a high level of reliability.

In the second and third groups, we performed certain degrees of rotation and scaling transformations on the initial images to evaluate the algorithms’ capability for image matching under different viewpoints and resolutions. The results indicate that both SIFT and SOSNet algorithms exhibit a significant decline in performance during the image matching process after rotation and scaling transformations, further highlighting their limitations. While D2-Net and SuperPoint algorithms also experienced a decrease in metrics, both maintained good overall performance with minimal differences between them. The multi-scale convolution aspect of the D2-Net algorithm provides advantages in scenarios involving scale variations. Additionally, the training process of the SuperPoint algorithm includes random rotation augmentation, which contributes to its robustness against angular rotations. LoFTR demonstrates good overall matching performance; however, it tends to lose features in scenarios with scale variations and incurs a higher computational cost. In contrast, our method achieves the best performance in terms of the number of correctly matched points (NCMs), success rate (SR), and root mean square error (RMSE), while also demonstrating excellent inference speed. Experimental results demonstrate that after descriptor enhancement, our method continues to deliver accurate and stable matching results in multi-temporal images with perspective and scale variations.

### 4.4. Ablation Study

To demonstrate the effectiveness of each module in the proposed method, we conducted ablation experiments to quantify the impact of each module on matching performance. With the pixel error threshold set to 3, and with other structures and processes kept constant, we randomly selected 10% of the experimental data from the initial image pairs in the test set for testing. This setup aims to enhance efficiency while validating the stability of the model design with fewer experimental data. We tested different combinations of these modules and compared whether the image matching framework utilized multiple perception layers and descriptor enhancement modules. The average quantitative results of these experiments are presented in Table 2.

It is evident from Table 2 that each module, when applied individually, demonstrated significant improvements relative to the baseline model. Specifically, removing the multiple perception layer led to a decline in the model’s feature point extraction performance, with a noticeable decrease in the NCM metric, as well as declines in SR and increases in RMSE. This indicates that the multiple perception layer can focus more effectively on prominent features while preserving important information in the images, thereby improving the quantity and positional accuracy of feature points. After removing the descriptor enhancement module, the model’s descriptor matching performance deteriorated significantly, with notable decreases in SR and increases in RMSE, alongside a slight decline in NCM. This is attributed to the descriptor enhancement module’s ability to enhance the geometric information of feature point descriptors while integrating global contextual information, which improves the model’s descriptor matching performance. Furthermore, the addition of both the multiple perception layer and the descriptor enhancement module increased the model’s network complexity to some extent, resulting in longer run times.

## 5. Conclusions

To address the challenge of matching remote sensing images in multi-temporal scenarios, in this paper, a method is proposed that enhances multi-temporal remote sensing image matching through multiple perception and descriptor augmentation. This approach integrates a feature extraction network informed by multiple perceptions to mitigate the poor matching outcomes caused by land cover changes over time and improve the extraction capability of effective feature points. Additionally, to improve matching performance for multi-temporal remote sensing images with varying angles and scales, a descriptor enhancement module comprising both self-enhancement and cross-enhancement stages is employed to optimize feature descriptors. The experimental results demonstrate that our method outperforms existing remote sensing image matching techniques in terms of accuracy and robustness on multi-temporal datasets.

However, the challenges of remote sensing image matching in multi-temporal contexts extend beyond land cover changes and variations in angle and scale; they also include sensor discrepancies and cross-modal image matching issues. The proposed method was primarily evaluated using homogeneous optical data in this study; thus, the effectiveness of its performance on highly heterogeneous or cross-modal data remains open to further investigation. Additionally, our experiments were conducted mainly on datasets covering specific geographic regions. Although the selected benchmarks are standard and widely utilized, our method’s performance requires comprehensive validation across different geographic areas and images with varying sensor characteristics. Therefore, future research should focus on these challenges to further enhance the practicality of the algorithm.

## Figures and Tables

**Figure 1 sensors-25-05581-f001:**
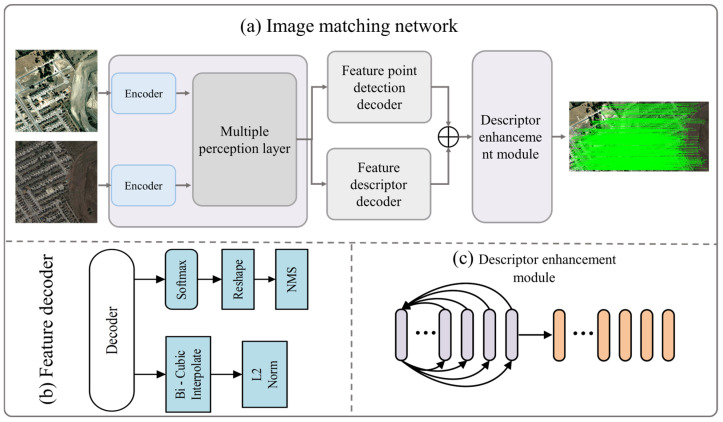
Workflow diagram of the image matching network.

**Figure 2 sensors-25-05581-f002:**
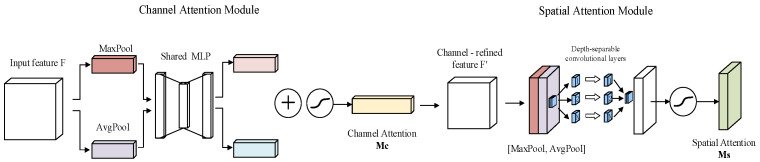
Structure diagram of multiple perception layers.

**Figure 3 sensors-25-05581-f003:**
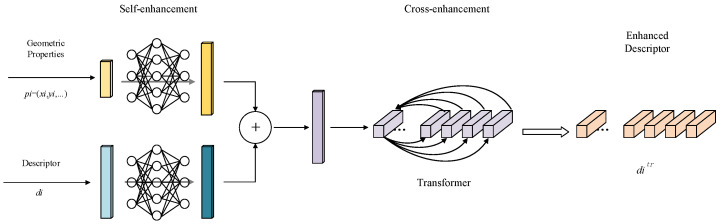
Structure diagram of the descriptor enhancement module.

**Figure 4 sensors-25-05581-f004:**
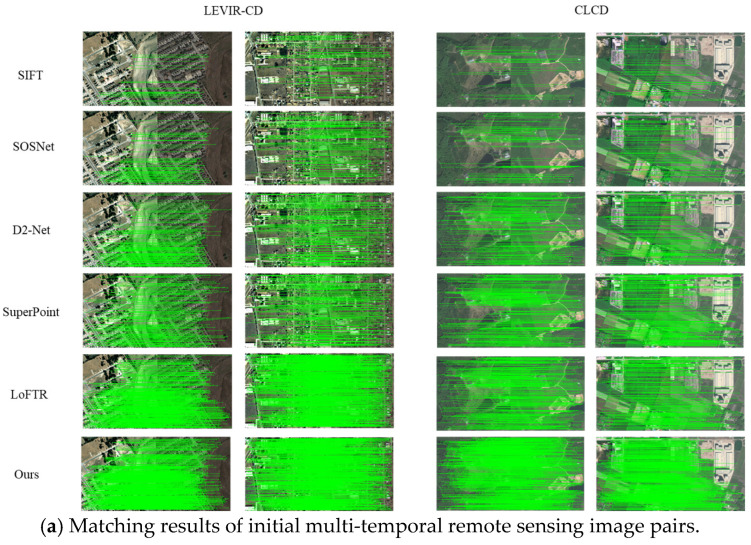
Visualization of partial matching results from the LEVIR-CD and CLCD datasets: (**a**) matching results of initial multi-temporal remote sensing image pairs; (**b**) matching results of multi-temporal remote sensing image pairs after 30° rotation; and (**c**) matching results of multi-temporal remote sensing image pairs after 0.7 scale reduction.

**Table 1 sensors-25-05581-t001:** Average matching results for all test data across various algorithms.

Group	Method	LEVIR-CD				CLCD			
		NCM	SR	RMSE	RT	NCM	SR	RMSE	RT
(a)	SIFT	82	86.74	2.34	2.53	76	86.53	2.52	2.61
	SOSNet	121	89.83	2.08	0.95	108	88.25	2.37	1.25
	D2-Net	160	91.42	2.17	1.47	177	93.36	2.06	1.37
	SuperPoint	173	92.46	1.88	0.76	156	90.14	2.25	0.92
	LoFTR	256	97.82	1.58	2.32	212	96.24	1.82	2.15
	Ours	263	99.25	1.62	1.24	246	98.43	1.71	1.30
(b)	SIFT	46	48.73	2.62	2.44	38	41.93	2.77	2.47
	SOSNet	81	55.16	2.58	1.02	66	45.00	2.61	1.18
	D2-Net	117	88.56	2.36	1.38	94	83.27	2.44	1.42
	SuperPoint	146	92.50	2.20	0.82	122	87.57	2.32	0.87
	LoFTR	187	93.38	1.98	2.17	165	90.29	2.04	2.24
	Ours	232	96.74	1.76	1.15	218	94.14	1.82	1.35
(c)	SIFT	52	53.60	2.50	2.23	43	47.81	2.95	2.55
	SOSNet	68	64.71	2.46	0.89	55	58.25	2.87	1.16
	D2-Net	133	91.54	2.28	1.31	120	88.05	2.39	1.33
	SuperPoint	110	86.21	2.41	0.85	102	82.34	2.65	0.78
	LoFTR	202	94.67	2.17	2.08	198	92.15	2.23	2.13
	Ours	218	97.48	1.67	1.21	203	95.86	1.74	1.28

**Table 2 sensors-25-05581-t002:** Ablation studies utilizing the test dataset.

Multi-Perspective Layer	Descriptor Enhancement Module	NCM	SR	RMSE	RT
×	×	202	91.56	2.13	0.86
√	×	243	94.83	1.97	1.14
×	√	217	96.71	1.72	0.98
√	√	256	98.24	1.65	1.27

## Data Availability

The LEVIR-CD dataset is available for download at: https://justchenhao.github.io/LEVIR/. The CLCD dataset is available for download at: https://github.com/liumency/CropLand-CD.

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
