# Peer review of "Multi-Temporal Remote Sensing Image Matching Based on Multi-Perception and Enhanced Feature Descriptors"

_sensors, 2025, doi:10.3390/s25175581_

Round 1
Reviewer 1 Report
Comments and Suggestions for Authors
Review of the article sensors-3818864
«Multi-Temporal Remote Sensing Image Matching Based on Multi-Perception and Enhanced Feature Descriptors»
Jinming Zhang, Wenqian Zang and Xiaomin Tian
The current stage of development of remote sensing methods is characterized by a huge accumulation of multitemporal data. multisensory data with different spatial resolution. The differences in the simultaneous images of various natural and artificial objects on the Earth's surface are due to many factors, from differences in the characteristics of reception equipment, illumination, geometric and weather conditions of the survey to changes in the objects themselves during the time between surveys. Comparative analysis of different time data obtained from different satellites is not a trivial task, but it is necessary to monitor and evaluate the dynamics of objects. Currently, the development of appropriate methods and technologies for remote sensing data processing is relevant. The authors of the reviewed article propose an image matching system based on multiple perception and an extended description of characteristics.
The basics of the proposed complex multilevel multicomponent method are similar to the principles of the SuperPoint and FeatureBooster methods and divide the processing process into two components: an optimized process for identifying characteristic points and improving object descriptors. The main processing steps include: 1) forming a tensor from high-level representations of objects by a feature extraction network from a set of input images; 2) determining the location of characteristic points of an object with several angles of view by the detection component; 3) generating descriptors for each point of objects; 4) independent and cross-improvement of descriptors and formation of extended descriptors. The originality and advantage of the technology developed by the authors lies in: 1) the approach of multiple information perception, which allows you to extract the most valuable information from each pixel of each channel; 2) the inclusion in the descriptors of the geometric information of the object itself and the global system of objects (within the image), which improves recognition accuracy, reliability, stability with respect to temporal changes, changing the angle of view and the complexity of the terrain; 3) maintaining high performance with the overall complexity of the architecture of this method through a convolution procedure with an eightfold reduction in image dimension.
The experimental verification of the method was performed on the basis of two data sets: 1) LEVIR-CD sourced from Google Earth with a spatial resolution of 0.5 meters from 637 pairs of images with a time interval of 5 to 14 years for built-up areas; 2) CLCD from the Gao Fen-2 satellite on the territory of cultivated lands of Guangdong Province, China, in 2017 and 2019, with a spatial resolution of 0.5 to 2 meters of 600 pairs of images. Some of the data is used for training, some for verification. In all cases, 4 indicators were used as performance indicators: the number of correctly matched points (NCM), matching success rate (SR), root mean square error of matched points (RMSE), and runtime (RT). To compare the efficiency of comparing multi-time images, the article also presents the results of data processing using 5 modern algorithms such as SIFT, SOSNet, D2-Net, SuperPoint and LoFTR. The results of processing the initial data demonstrate the highest rates of NCM, SR (SR=99.25% for built-up areas and SR=98.43% for cultivated land) of the presented method out of 6 methods, according to RMSE, this method is slightly inferior to the LoFTR method, and in terms of processing speed, SuperPoint and SOSNet methods. The analysis of changes in the efficiency and performance indicators of all 6 methods was also carried out when images were rotated by 30° and when the scale was changed by 0.7 times. For the method presented in the article, there is complete preservation of functionality and a slight deterioration in efficiency and productivity indicators.
The technology presented by the authors for comparing multi-time remote sensing data is innovative and is of great interest to researchers in this field. It is recommended for publication after a little revision.
Remarks:
- In line 74, the phrase "In summary" seems inappropriate for the introductory part.
- In clause 3.1, the description in lines 121-131 does not fully correspond to the blocks of the processing process diagram in Fig.1. At the end of paragraph 3.1, add a brief description of the intended product or processing result (what is the output?).
- The description of the Shared Encoder in lines 136-140 is not very clear; it is necessary to give a clearer idea or a link to a literary source with such.
- The phrase "scalar vectors" in line 150 is contradictory, it is necessary to clarify or replace it with another term.
- After formulas (1)-(4), (7) It is necessary to provide an explanation of the designations of all variables and abbreviations.
- When describing the experimental data sets in clause 4.1, the total number and amount of data for training and validation do not match. For example, for LEVIR-CD, the total number of pairs is 637, of which 445 are for training, 64 for verification. 445+64=509, and what about the other 637-509=128 pairs? A similar discrepancy exists for the CLCD dataset.
- It is advisable to provide a justification or at least a hint of the reason for choosing the values of the rotation angle and zoom level, presented in lines 279-280.

Reviewer 2 Report
Comments and Suggestions for Authors
This paper presents a remote sensing image matching approach with multi-temporal and multi-perception based on improved feature descriptors. Although the paper deals with an important issue of remote sensing image processing, there are some key issues to be resolved prior to publication recommendation.
Specific Line-by-Line Comments
- Line 30-31: The phrase "disaster assessmen" contains a typographical error - should be "disaster assessment".
- Lines 53-73: The introduction confesses to having borrowed ideas from FeatureBooster and SuperPoint but does not provide adequate distinction to determine if the approach presented is new compared to merely combining established methods.
- Lines 74-83: The contributions claimed are exaggerated. Contribution 1 guarantees to "enhance both the quantity and quality of feature points" but is not backed by theory as to why the multi-perception strategy would perform better than established multi-scale methods.
- Lines 132-140: The shared encoder description is too superficial. Critical details are missing: Specific layer configurations. Number of filters per layer. Activation functions used. Parameter initialization strategies
- Lines 154-155: Equation (1) requires clarification: 𝑀𝑐(𝐹) = 𝜎(𝑀𝐿𝑃(𝐴𝑣𝑔𝑃𝑜𝑜𝑙(𝐹)) + (𝑀𝐿𝑃(𝑀𝑎𝑥𝑃𝑜𝑜𝑙(𝐹))))
The MLP architecture specifications are missing. What are the hidden layer dimensions, activation functions, and dropout rates? - Lines 162-163: Equation (2) uses unclear notation: 𝑀𝑠(𝐹) = 𝜎(𝑓([𝐴𝑣𝑔𝑃𝑜𝑜𝑙(𝐹); 𝑀𝑎𝑥𝑃𝑜𝑜𝑙(𝐹)]))
The concatenation operator "[;]" needs explicit definition regarding dimension and the function f requires specification. - Lines 165-167: Equations (3) and (4): 𝐹′ = 𝑀𝑐(𝐹) ⊗𝐹, 𝐹" = 𝑀𝑠(𝐹′) ⊗𝐹′
The element-wise multiplication operator ⊗ should specify broadcasting behavior and dimensional compatibility constraints. - Lines 177-189: Key point detection decoder description lacks mathematical formulation. How is the 65th "invalid channel" being included in the softmax calculation? What are the actual NMS parameters and threshold values?
- Lines 202-203: Equation (5): 𝑑𝑖𝑡𝑟←𝑀𝐿𝑃𝑑𝑒𝑠𝑐(𝑑𝑖)
The MLPdesc model is fully specified such as layer size, activation functions, and regularization methods. - Lines 209-210: Equation (6): 𝑑𝑖𝑡𝑟←𝑑𝑖𝑡𝑟+ 𝑀𝐿𝑃𝑔𝑒𝑜(𝑝𝑖)
Geometric embedding and descriptor dimension compatibility are not checked. How is the addition operation guaranteed? - Lines 224-225: Equation (7): (𝑑1𝑡𝑟, 𝑑2𝑡𝑟, … 𝑑𝑁𝑡𝑟) ←Trans(𝑑1𝑡𝑟, 𝑑2𝑡𝑟, … 𝑑𝑁𝑡𝑟)
The Transformer architecture is completely unspecified. Critical missing details include: Number of attention heads. Embedding dimensions. Feed-forward network architecture. Positional encoding scheme. Number of layers - Lines 233-242: Dataset descriptions are incomplete: Ground truth generation strategy. Annotation quality assessment. Potential dataset biases. Cross-validation procedures.
- Lines 243-246: Experimental setting is not complete: No mention of random seed for reproducibility. No mention of learning rate scheduling details. No mention of convergence criteria. No mention of early stopping methods.
- Lines 255-267: Evaluation metrics (Equations 8-10): 𝑁𝐶𝑀= ∑ᵢ₌₁ⁿ 𝕀(∥𝑥ᵢ−𝑥̅ᵢ∥≤𝜀), 𝑆𝑅=𝑁𝐶𝑀/𝑁 , 𝑅𝑀𝑆𝐸=1/𝑁𝐶𝑀 ∑ᵢ∥𝐻(𝑥ᵢ)−𝑥̅ᵢ∥. These metrics lack statistical rigor. Missing elements include: Confidence intervals. Statistical significance testing. Error bar calculations. Cross-validation results.
- Lines 270-283: Statistical confirmation of experimental comparison is not proven. Results must include: Multiple runs with various random seeds. Statistical significance tests (t-tests, ANOVA). All measures accompanied with confidence intervals.
- Lines 318-319: "Our method had the maximum value for correctly matched points (NCM) and SR, with 263 and 99.25%" needs statistical proof to verify if improvement is statistically significant.
- Lines 356-357 Table 2 offers ablation results but is missing: Statistical testing of significance. Experimental runs in multiples. Confidence intervals. Analysis of interaction effects among modules.
- Missing Critical Elements
- Loss Function: No mathematical formulation of the training objective is provided.
- Computational Complexity: No analysis of time/space complexity or scalability assessment.
- Failure Analysis: No discussion of method limitations or failure cases.
- Cross-domain Validation: Testing limited to two similar datasets without geographic or sensor diversity.
- Reproducibility: Missing critical implementation details for reproducibility.
- Reference Analysis and Suggestions: The reference list contains 34 citations with good coverage of recent work (80.8% from 2020-2025). However, several important areas are underrepresented. Missing Recent Work on: Transformer-based remote sensing applications (2023-2025). Uncertainty quantification in deep learning. Multi-temporal change detection methodologies. I would suggest the following references: https://doi.org/10.3390/opt5040029 and https://doi.org/10.5152/forestist.2025.24068.
- Major Concerns
- Limited Novelty: The method considerably uses past methods with comparatively low theoretical innovation.
- Mathematical Rigor: Inadequate definitions of network structures and non-formulation of loss functions.
- Statistical Validation: Lack of statistical significance testing and confidence intervals.
- Reproducibility: Insufficient implementation details for independent verification.
- Limitations of evaluation: Testing only on two datasets without cross-domain evaluation.
- Recommendations
- Introduction: Needs improvement - Enhance motivation and distinguish from previous work effectively
- Research Design: Needs improvement - Offer better theoretical support for multi-perception approach
- Methods: Need to be enhanced - Full mathematical specifications and architectural details
- Results: Can be improved - Include statistical verification and test for significance
- Conclusions: Can be improved - Moderate claims based on evidence presented
Round 2
Reviewer 2 Report
Comments and Suggestions for Authors
The revised draft shows substantial improvement compared to the original. The majority of the problems listed in the first round (e.g., insufficient methodological detail, incomplete equations, incomplete descriptions of architectures, missing clarity regarding datasets, and incomplete loss function information) have been properly addressed. Mathematical formulations were contributed by the authors, encoder-decoder architecture was explained, the description of MLP and Transformer modules was enhanced, and data set use and test settings were discussed. These contributed towards making the technical readability of the manuscript improved. Some of the remaining issues that need to be considered that help in improving the manuscript to come out in better shape:
- While some reviewers included additional clarifications, experiment results are still largely presented in averages without confidence intervals or statistical significance tests. Improved argumentation of reproducibility would make the statements even more robust.
- Generally clean language but still contains some unnatural phrasing and slight grammatical errors. Professional language editing for readability would enhance it.
- The paper would be better served by a brief reference to limitations (e.g., computational expense, scalability, or possible failure modes in highly heterogeneous or cross-sensor data). This would strike a balance and allow readers to estimate applicability more easily.
- While LEVIR-CD and CLCD datasets are suitable, both are for Chinese regions. Testing on datasets with varied geographic/sensor properties would enhance generalizability even more. At least, do so as a limitation.
